# *k*-NDDP: An Efficient Anonymization Model for Social Network Data Release

**Shafaq Shakeel [1], Adeel Anjum [1,2] , Alia Asheralieva [2,\*] and Masoom Alam [1]**

[1]   Department of Computer Sciences, Comsats University, Islamabad 44000, Pakistan;
     shafaqshakeel555@gmail.com (S.S.); adeel.anjum@comsats.edu.pk (A.A.);
     masoom.alam@comsats.edu.pk (M.A.)
[2]   Department of Computer Science and Engineering, Southern University of Science and Technology,
     Shenzhen 518055, China
[\*]   Correspondence: asheralievaa@sustech.edu.cn

**Abstract:** With the evolution of Internet technology, social networking sites have gained a lot of popularity. People make new friends, share their interests, experiences in life, etc. With these activities on social sites, people generate a vast amount of data that is analyzed by third parties for various purposes. As such, publishing social data without protecting an individual's private or confidential information can be dangerous. To provide privacy protection, this paper proposes a new degree anonymization approach *k*-NDDP, which extends the concept of *k*-anonymity and differential privacy based on Node DP for vertex degrees. In particular, this paper considers identity disclosures on social data. If the adversary efficiently obtains background knowledge about the victim's degree and neighbor connections, it can re-identify its victim from the social data even if the user's identity is removed. The contribution of this paper is twofold. First, a simple and, at the same time, effective method *k*–NDDP is proposed. The method is the extension of *k*-NMF, i.e., the state-of-the-art method to protect against mutual friend attack, to defend against identity disclosures by adding noise to the social data. Second, the achieved privacy using the concept of differential privacy is evaluated. An extensive empirical study shows that for different values of *k*, the divergence produced by *k*-NDDP for CC, BW and APL is not more than 0.8%, also added dummy links are 60% less, as compared to *k*-NMF approach, thereby it validates that the proposed *k*-NDDP approach provides strong privacy while maintaining the usefulness of data.

**Keywords:** degree anonymity; *k*-anonymity; mutual friend attack; *k*-NMF; differential privacy

## 1. Introduction

Social networking sites have gained popularity due to the advanced features they provide to the users. Because of this popularity, people use social sites to connect with their friends and family, share their interests, and establish connections. The information about the user's interest and the connection is publicly available to everyone, although a user can change their privacy settings and set the differential access to their private information [1]. By using the social networking sites, a user generates a massive amount of data daily. Social network service providers collect and maintain this generated data. Generated data is being published by network providers for research purposes which is useful in domains like marketing and survey, as shown in Figure 1. While making data publicly available, protection of the sensitive information becomes of the prime concern. In particular, social data contains a lot of important information, such as user contact details that must not be disclosed. Social data is typically published in the form of social graphs where each vertex is a representation of an individual and their links represent their connections. The problem with such types of data publishing is that the adversary easily gets the victim's connection information (background knowledge) from their social media account and uses it to identify its victim on a published graph [2]. If the adversary successfully identifies its victim from a

social graph, we say privacy is breached. A privacy violation [3] is an action that exposes private or confidential information about an individual to the adversary. One common approach to provide defense against privacy breaches is simply removing user identities from a published graph [4]. However, this solution is not always appropriate due to the availability of useful information on different social sites. Adversaries intelligently apply different queries to get background knowledge about a user, such as number of connections, mutual connections, and connections among connections, etc.

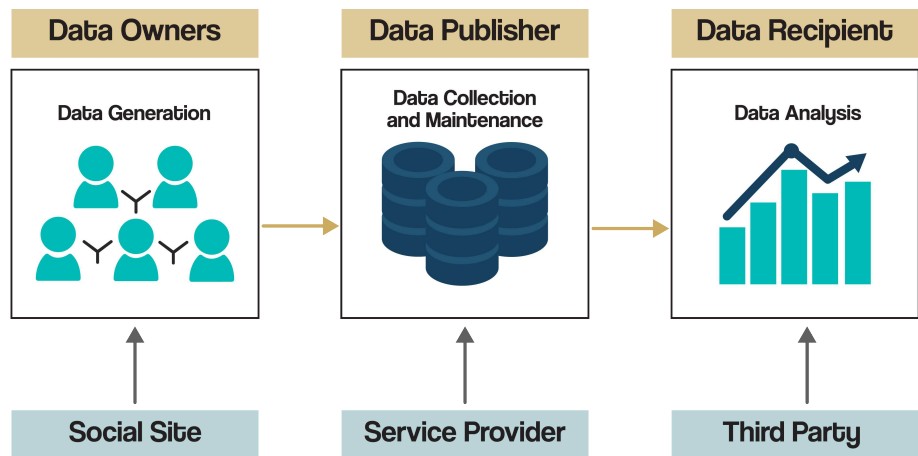

**Figure 1.** Social networks data publishing scenario.

For privacy preservation, the social graph should be anonymized before releasing it for research purposes. For privacy persevered data publishing, many models are proposed for relational and social datasets.

### 1.1. Motivational Example

With the background knowledge, the adversary tries to obtain sensitive information about the victim using a published dataset [5]. Based on this background information, the adversary performs three different types of attacks on a published data. These attacks are categorized as identity disclosure, sensitive attribute disclosure, and sensitive link disclosure. Identity disclosure [6] is a type of breach that occurs when an adversary reveals the individual behind the record. Sensitive attribute disclosure collects sensitive or confidential details about the individual [7]. Correlation between different sensitive attributes might cause identity disclosure. Sensitive link disclosure discovers the relation between two individuals. It causes a mutual friend attack [8]. Among all of these privacy issues, identity disclosure is most of concern.

To elaborate accurately, consider the example of a graph shown in Figure 2, where each vertex represents a person and their links represent their connections. Suppose Figure 2b is the published graph. It is being published after applying sufficient privacy by anonymizing its vertices. Before initiating any attack, the adversary collects background knowledge about its victim. Now, presume that adversary obtains information about its victim's neighbor connections (Degree of a vertex) and connection among neighbors (neighbor mutual connections). Once the adversary effectively obtains such information, it discerns the position of its victim from the graph structure because of uniqueness, and it can certainly identify the individual.

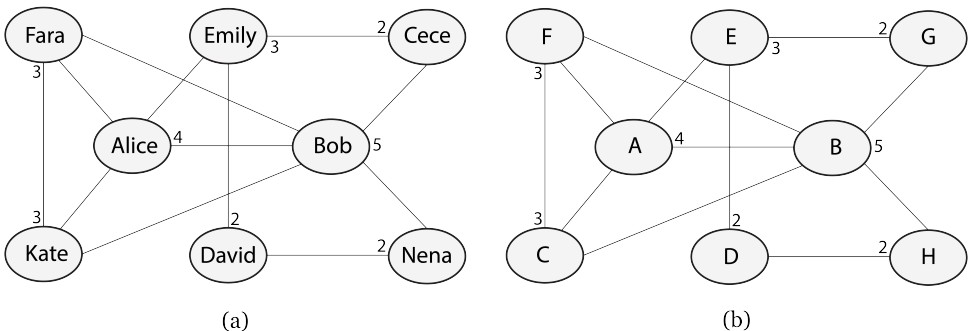

**Figure 2.** Social graphs: (**a**) Original graph, (**b**) naïve anonymized graph.

Suppose the adversary knows Bob has five friends (Alice, Fara, Cece, Kate, and Nena) and some of them know each other; say Fara and Kate and Emily and Alice are friends. Figure 3a shows the neighborhood graph of Bob. Unfortunately, Bob can be recognized from the published social network graph, as no other vertex has the same graph structure as Bob, which causes identity disclosure. Similarly, an adversary can identify any individual whose neighborhood graph is available to it. If the adversary identifies both Alice and Bob from the graph, it might perceive that both are friends and they share the same mutual friends, which causes sensitive link disclosure. To provide defense against such intrusions, inserting dummy links in a graph is one possible and highly accepted solution. So, the probability to identify a user from an anonymous social graph is less or equal to $1/k$. This concept is proposed by L.Sweeney, called $k$-anonymity [9,10]. Which guarantees anonymization of individuals on published datasets [3].

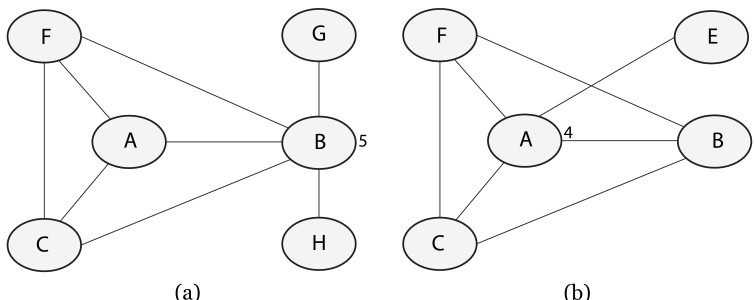

**Figure 3.** Neighborhood graphs in a social network: (**a**) Graph for Bob, (**b**) graph for Alice.

### 1.2. Challenges

Privacy preservation is widely considered an important topic for data publishing, and many models are proposed as well as many effective algorithms. However, these methods cannot be directly applied to social graphs as they are ideally proposed for relational datasets and graph anonymization is more complicated than relational data [6]. The reasons are listed below:

- To breach an individual's privacy, the adversary uses a variety of background knowledge. Modeling their background knowledge, capabilities, and types of attacks it might perform on released social data is a complicated task [11]. The reason is that every bit of detail on graphs can be used to perform any attack, such as the label of vertices, no links, etc. Thus, it is very difficult and complicated to find a suitable privacy model for social networks.
- The relational data contains the list of tuples where each tuple is independent of others. Applying any anonymization method on the set of rows does not influence other rows in the relational datasets [12]. The social graph is a correlation of vertices and links, any change can affect the whole graph structure. So, it is much more challenging

to choose an appropriate graph anonymization method and maintain the balance between privacy and utility [13].
- The third issue is related to data utility. Anonymization of graphs is much more different and complicated than relational datasets. Data loss is evaluated using tuples in relational datasets, whereas in social graphs it is rather different. The reason is that social data is a combination of vertices and links. Unlike relational datasets, we cannot correlate social graphs even if they have the same numbers of vertices and links [14]. Every graph is different in terms of APL, CC, and BW.

### 1.3. Contribution and Organization

Social network data publishing while maintaining privacy and utility is a challenge that cannot be solved in one shot. In this paper, the concept of *k*-NDDP is presented, which provides defense from identity disclosure that intrudes privacy of individual by discerning its position on social graph. The key contributions of this paper are following:

- It proposes the concept of *k*–NDDP(*k*-Node Degree Differential Privacy) which injects noise for synthesizing the social graph and defends against identity disclosures. Then design and implementation of *k*-NDDP algorithm is done where the degrees are partitioned into k-anonymized groups such that degree anonymization cost is minimized and the whole social graph is reconstructed using anonymous degree sequences. The anonymized graph protects the vertices and their associated links.
- For privacy analysis, it uses the concept of differential privacy to analyze the *k*-NDDP in terms of privacy preservation. Furthermore, evaluates the efficiency of *k*-NDDP on three real-life datasets. Empirical study indicates that proposed method outputs the graph which correctly preserves the structural properties of the original graph also provides strong privacy.

The organization of the paper is as follows. Existing possible methods for social graph anonymization are described in Section 2. The practical solution of a degree anonymization method is described in Sections 3 and 4, respectively. The proposed method is examined empirically using real data-sets in Section 5. Finally, Section 6 concludes this research.

### 2. Related Work

Social networking sites provide many attractive features to their users. People use these social sites for different purposes including sharing their data publicly with friends. Generated social data has huge commercial value as well as also containing sensitive information that needs to be protected. Social data should be properly anonymized to maintain the user's data privacy before publishing. The adversary uses a range of background information about the target individual to infer its private information [15]. Background knowledge is referred to as information an adversary uses to perform any attack on a published dataset [16]. Many privacy-preserving techniques are introduced for social data publishing. These techniques are categorized as Graph Modification, Clustering, and Differential Privacy based techniques, which are shown in Figure 4. The main objective of privacy-preserving techniques is to conceal confidential details of individuals in the published dataset while maintaining its usefulness [17].

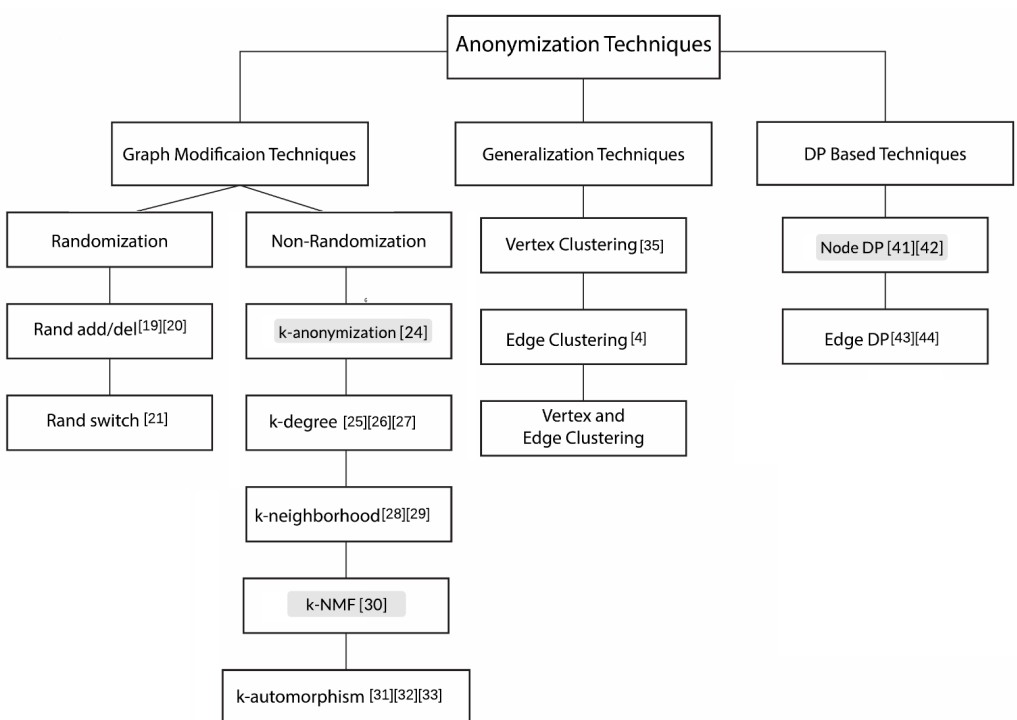

**Figure 4.** Anonymization techniques for social networks data publishing.

### 2.1. Graph Modification Techniques

These methods anonymize the social network by modifying its vertices or links. Graph Modification Method is further categorized as randomization and non-randomization method. In the randomization method, the whole graph is anonymized randomly by inserting noise. This technique removes identifiable information of individuals by inserting or deleting links or vertices [18]. This addition or deletion is made randomly, thus this technique is called randomization or random perturbation. To change network structure, two basic link modification techniques are proposed. Rand add/del: this technique randomly inserts one link and deletes another link from the graph and thus preserves the total no of links [19,20]. Rand switch: this technique selects two links randomly and makes new connections by exchanging their vertices [21]. New connections do not exist in the original graph. Non-randomization techniques require the addition or deletion of links/vertices to meet some desired conditions [22]. Recent studies on non-randomization techniques prove that they accomplish a significant level of anonymization while maintaining the original structure of a graph [23]. Some famous non-randomization techniques are listed below in Table 1:

- *k*-anonymization: *k*-anonymization method modifies the original graph by inserting/deleting links or vertices to obtain certain requirement [24]. If an adversary has background knowledge of degrees and relationship details about its victim, it tries to re-identify the victim from the anonymized graph. Different *k*-anonymity based anonymization methods differ by the type of adversary's background knowledge.
- *k*-degree anonymity: Degree-based anonymization methods provide prevention from attacks when the adversary uses degree information about the target as background knowledge to identify its victim from a naive anonymized graph. *k*-degree anonymity transforms the original graph into the new modified version by only adding vertices or links or both [25]. The main objective is to insert the minimum number of vertices or links to maintain the characteristics of the original graph. Ref. [26] proposed *k*-degree-based method, using a dynamic anonymization process. The method in [27] modifies the original graph into *k*-degree-based anonymized graph by adding fake vertices rather than new links.

- *k*-neighborhood anonymity: The adversary uses background knowledge of its victim's degree and neighbor connections to perform neighborhood attack and *k*-neighborhood anonymity is used to provide the defense [28]. Ref. [29] proposed an algorithm to provide defense from neighborhood attacks by inserting links or vertices in the original graph until there are *k* vertices and their subgraphs becomes isomorphic. Ref. [30] identified an attack called mutual friend attack that occurs due to disclosure of sensitive link between two vertices. Ref. [31] proposed an algorithm (*k*-NMF) that adds extra links into the original graph and ensures that there must be at least $k - 1$ other links with the same number of mutual friends.
- *k*-automorphism: The adversary uses background knowledge of a subgraph to identify the target from a published graph [32]. Ref. [33] proposed a method to prevent subgraph-based privacy attacks. In Ref. [34], the method modifies the original graph by providing *k* identical subgraphs where every vertex is automorphic with other $k - 1$ vertices. *k*-automorphism provides strong security against structural subgraph attacks [35].

**Table 1.** Graph anonymization techniques.

| Ref. No. | Technique | Anonymization Strategy | Advantages | Limitations |
|---|---|---|---|---|
| [24] | *k*-anonymization | Add or deletes links/vertices | Protects link identity | High algorithm complexity |
| [19] | Rand add/del | Randomly adds or deletes links/vertices | Maintain the actual number of links | Not considered any adversarial attack |
| [36] | Vertex Clustering | Vertex clustering | Allow structural queries from different domains | Difficult to analyze local structural details in the graph |
| [4] | Edge Clustering | Vertex generalization | Balances utility and privacy | Execution time increases with generalization |
| [37] | Edge Differential Privacy | Link addition or deletion | Protect relation between two vertices | Susceptible to vertex identification |
| [38] | Node Differential Privacy | Vertex modification | Protect vertex and adjacent links | N/A |
| [25] | *k*-degree anonymity | Addition of fake links/vertices | Conserve much of the characteristics of the graph. Prevent vertex identification problem | Unsecured neighbor connections |
| [31] | *k*-NMF | Fake links addition | Protect sensitive links between vertices | Unsecured neighbor connections and Increased runtime |
| [34] | *k*-automorphism | Vertex modification | Strong security against structural subgraph attacks | Susceptible to identity disclosure |
| [21] | Rand switch | Switch old links with new links | Preserve spectral characteristics of the graph | Not considered any adversarial attack |
| [29] | *k*-neighborhood anonymity | Fake vertices and links addition | Re-identification attack protection | Utility loss (extreme change) |

### 2.2. Generalization Techniques

This technique is also called the clustering-based approach. The main theme of generalization techniques is to group the vertices or links into a cluster and then form a super vertex [23]. Each super vertex contains merged details of the sub-network. This approach

makes it difficult to analyze the local structural information of the graph. Ref. [36] proposed an approach for grouping the vertices with similar structural properties, where no of vertices in each cluster must be $\geq k$. Ref. [4] proposed vertex generalization method where vertices of the graph are grouped as disjoint sets. This approach maximizes data utility.

### 2.3. DP Based Techniques

Background knowledge is not considered in Differential Privacy [39–42]. As it provides the guarantee that the adversary will not be able to determine any sensitive information about its victim. Differential privacy is interactive and non-interactive. Both ways protect user's private information by adding noise on either query results or in actual data. The common method for noise addition is Laplace distribution [43]. Differential privacy works on two different datasets, say $G_1$ and $G_2$ both differing on at most one vertex/link. A basic definition of differential privacy says that the probability of output $O$ by applying function $Al$ on $G_1$ is at most the same as the probability of output $O$ by applying function $Al$ on $G_2$ even when both datasets differ by one element.

$$Pr[Al(G_1) = O] \leq e^{\epsilon} \times Pr[Al(G_2) = O]$$

Parameter $\epsilon$ should be chosen carefully, as it controls privacy and utility trade-off. The lesser the value of $\epsilon$, the more privacy is achieved with lower accuracy, and vice versa [41]. Differential privacy is further categorized into two groups. Node and Edge DP.

- Node Differential Privacy: Node Differential Privacy protects a vertex and its associated links. In Node Differential Privacy, social graphs $G_1$ and $G_2$ (which are obtained by adding or removing vertices and their associated links) are said to be vertex neighbors of each other Ref. [44]. Different approaches are developed to realize vertex differential privacy. Ref. [38] proposed many node differential privacy algorithms and methods to analyze the accuracy of those algorithms. Ref. [45] proposed a new concept to achieve vertex differential privacy and discussed some problems to accomplish it.
- Edge Differential Privacy: Edge Differential Privacy only protects connections between two vertices where two graphs $G_1$ and $G_2$ are link neighbors of each other. Ref. [46] presented Edge Differential Privacy technique for general subgraphs. Ref. [37] proposed an Edge DP for the case of spanning trees and triangle problems in a graph.

The main objective of this paper is to publish an anonymized social graph and protect individuals from identity disclosures. As the adversary uses degree and neighbor connections information about the target to discern the position of its victim, both need to be protected or anonymized. To achieve this goal, this paper proposes the approach $k$–NDDP, which is the extension of $k$-NMF [31]. The proposed technique inserts the minimum number of dummy links into the original graph before publishing and defends against identity disclosures.

### 3. Preliminaries

Our social network graph is an undirected graph $G = (V, L)$, where $V$ represents the set of $(v_1, v_2, \dots)$ and $L$ represents the set of links between the vertices.

### 3.1. Degree Sequence Partitioning (DSeq)

A social graph, consisting of multiple vertices and links $G = (V, L)$, where vertices are defined as $V = (v_1, v_2, \dots, v_n)$ and $L$ is the link between them. *DSeq* is a vector that contains the degree of all vertices in descending order.

### 3.2. Degree Anonymization

Given the *DSeq* of original graph $G = (V, L)$, and anonymization parameter $k$, construct $k$-anonymous $DSeq_a$ such that degree anonymization cost is minimized.

### 3.3. Graph Reconstruction

Given *k*-anonymous $DSeq_a$, construct anonymized graph $Ga(V_a, L_a)$ such that $|L_a| \cap |L| = |L|$ and graph anonymization cost is minimized.

### 3.4. Differential Privacy

A randomized algorithm *Al* satisfies DP for any given graph $G_1$ and its neighboring graph $G_2$ for any possible output $O$ for *Al*. $Pr(.)$ is the probability of having the same output. $\epsilon$ is the privacy budget that controls the *Al*. $Pr[Al(G_1) = O] \leq e^\epsilon \times Pr[Al(G_2) = O]$

### 3.5. Node DP

A randomized algorithm *Al* satisfies DP for any given graph $G_1 = (V_1, L_1)$ and its neighboring graph $G_2 = (V_2, L_2)$ where $V_2 = V_1 - a$ and $L_2 = L_1 - (v_1, v_2)$ where $v_1 = a$ OR $v_2 = a$ for some $a \in V_1$.

### 3.6. Identity Disclosure

For a social graph $G = (V, L)$, it is defined as background knowledge about $DoV$ and neighbor mutual connections for vertex $V$. identity disclosure discerns the position of vertex such that $v \in Va$ from $Ga$.

### 3.7. Mutual Friend Attack

For a social graph $G = (V, L)$, it is defined as background knowledge of common connections between two vertices. Common connections between $v_1$ and $v_2$ are their mutual friends.

### 3.8. k-NMF

For a social graph $G = (V, L)$, it is defined as common neighbor vertices between two end vertices. For privacy parameter *k*, there is at least *k*–1 number of other links that have the same NMF value shown in Figure 5.

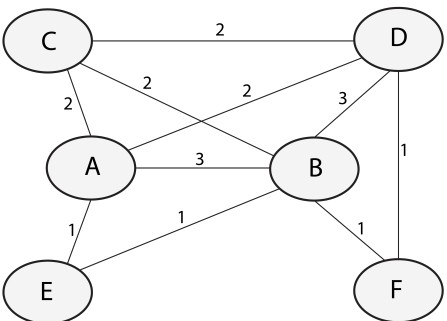

**Figure 5.** *k*-NMF anonymized graph.

## 4. The Proposed *k*-NDDP Approach

In this paper, a degree anonymization technique called *k*-NDDP is proposed for undirected social graphs. The technique takes input as an original graph $G(V, L)$ and anonymization parameter *k* to generate anonymized graph $G_a(V_a, L_a)$. For graph modification, it uses link insertion operation, which adds a minimum no of dummy links into the original graph and minimizes anonymization cost, and preserves structural properties between $G$ and $G_a$. The technique consists of three steps and proceed as follows:

1. First, it creates the sequence partitioning ($DSeq$) of $G$ in descending order and constructs new degree sequence partitioning ($DSeq_a$) that is *k*-anonymous and such that degree partitioning cost is minimized.

$$DA_{cost}(DSeq_a, DSeq) = Dis(DSeq_a - DSeq)$$

2. Given new degree sequence partitioning $DSeq_a$, it constructs a new anonymized graph $G_a(V_a, L_a)$ such that $|L_a| \cap |L| = |L|$ and graph anonymization cost is minimized.

$$G_{cost} = (G_a, G) = |L_a| - |L|$$

3. Lastly, it uses the concept of differential privacy to analyze the proposed approach in terms of privacy protection.

### 4.1. Proposed Methodology

To prevent identification attacks, different privacy-preserving techniques for social networks are proposed, but the concept of k-anonymity is widely accepted. This paper proposes the concept of *k*-NDDP for degree anonymization of social graphs to provide defense against identity disclosures. In the proposed solution, the concept of *k*-anonymity is extended for vertex degree and it specifically prevents identification attack by adding dummy links into the original graph. The proposed technique anonymizes the social graph such that there must be at least k vertices with the same degree or graph topology as the victim vertex. The first step is datasets cleaning. As undirected and unweighted social graphs are used for the experiments, there is a possibility of incomplete vertices available in the graphs, which are removed during the data cleaning process. The next step is partitioning of degree sequence in descending order and anonymize the sequence to reconstruct the graph. The proposed *k*-NDDP anonymization process for social data is presented in Figure 6.

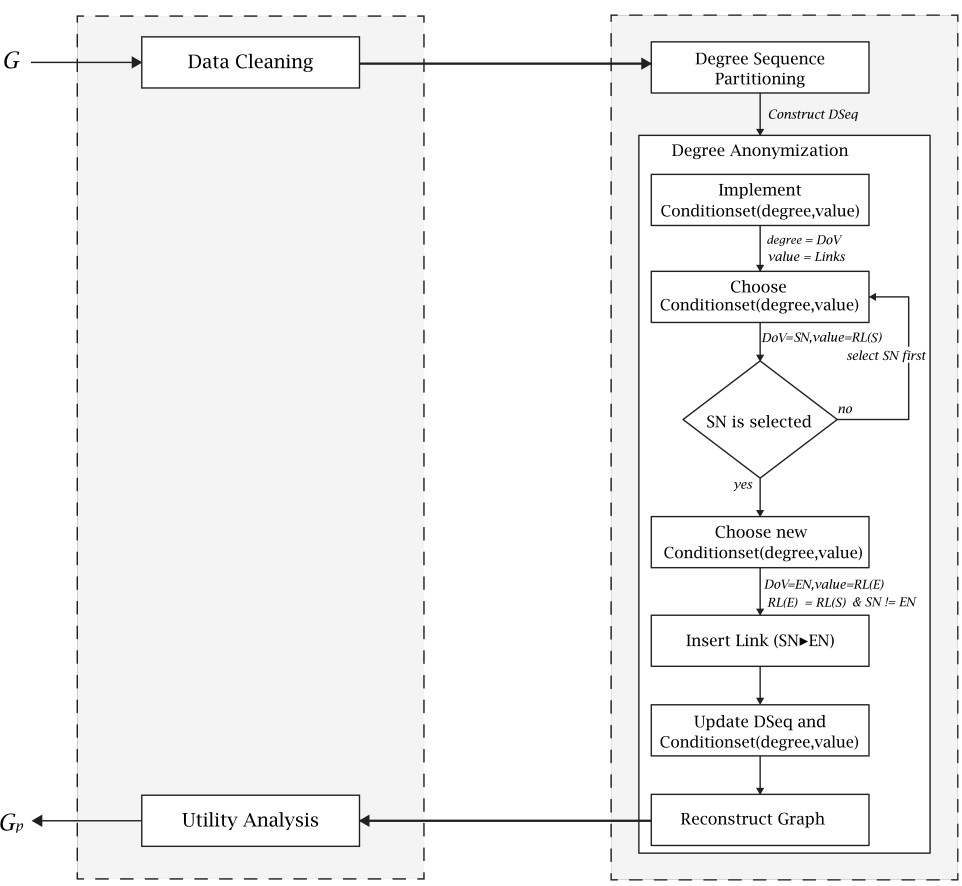

**Figure 6.** Overview of proposed *k*-NDDP approach.

4.1.1. Degree Sequence Partitioning (*DSeq*)

The goal of this step is to find candidate vertices and their required number of links for link insertion operation. Degrees of vertices are arranged in descending order and then partitioned into different groups shown in Algorithm 1.

---

**Algorithm 1:** Degree sequence partitioning

1: DSeq = []
2: **for each** link($v_1, v_2$) **do**
3:　　cnt = DSeq_val($v_1, v_2$)
4:　　insert cnt in DSeq in descending order
5: **end for**
6: last_index = 0
7: **for** $x = k$ to *total_links* $- k$ **do**
8:　　$C_1 = $ DSeq [last_index] $-$ DSeq [$x$]
9:　　$C_2 = 0$
10:　　**for** $y = x + 1$ to $1 + k$ **do**
11:　　　$C_1 = C_1 +$ DSeq [$x + 1$] $-$ DSeq [$x$]
12:　　　$C_2 = C_2 +$ DSeq [$x$] $-$ DSeq [$y - 1$]
13:　　**end for**
14:　　**if** $C_2 < C_1$ **then**
15:　　　last_index $= x$
16:　　　$x = x + k$
17:　　**else**
18:　　　$x + +$
19:　　**end if**
20: **end for**

---

For partitioning, the proposed solution uses the greedy algorithm, which first makes the group of $k$-highest degree elements and then checks whether to merge the next $(k+1)$th element into the existing group or start a new group from $(k+1)$th element. This decision depends on two different partitioning costs.

$$C_1(Integration\ Cost) = [D(k+1) + D(next\ k\ elements)]$$

$$C_2(New\ Group) = [D(k\ elements)]$$

If $C_1$ is greater than $C_2$, a new group starts from $(k+1)$th element. Furthermore, if $C_1$ is less than $C_2$, $(k+1)$th element is merged into the previous group and the new group starts from $(k+2)$ element. This algorithm proceeds recursively for the sequence $D[K+1, n]$ and terminates after considering all $n$ vertices. For every $i$ element of *DSeq* of size $n$, the greedy algorithm checks other $O(k)$ elements to decide whether to merge the next element into the previous group or to start a new group. As there is $n$ number of elements (vertices), so the total running time of this algorithm is equal to $O(nk)$.

4.1.2. Degree Anonymization and Graph Reconstruction

After degree sequence partitioning, the next step is to implement the *conditionset* (*degree, value*) shown in Algorithm 2. All elements of the *DSeq* have an entry in the *conditionset*(), which represents the degree of candidate vertices and their required links for degree anonymization. Consider $E$ is the element of a group that belongs to *DSeq*, and $C_{D(E)}$ is the current degree of that element and $T_D$ is the target degree to be achieved (Degree of the first element). For each non-zero value of all groups in *DSeq*, we compute $[C_{D(E)}, T_D - C_{D(E)}]$ and make an entry as (*degree, value*) pair in the condition set. After condition set implementation, we select candidate vertices for link insertion operation. Candidate selection is an iterative process. In the first iteration, the algorithm chooses the first pair from *conditionset*(*degree, value*) and assigns its degree to $S_N$ (starting node) and value to $R_{L(S)}$ (required links). During the second iteration, the algorithm finds the pair which

requires the same links as $R_{L(S)}$ such that $R_{L(S)} = R_{L(E)}$. After successful pair selection, the new (*degree*, *value*) pair is assigned to $E_N$ (ending node) and $R_{L(E)}$ (required links), respectively. Once $S_N$ and $E_N$ are selected, the algorithm selects the optimal vertices from the graph for new link insertion. In social graphs, there is a possibility of more than one vertices having the same degrees. To select the optimal vertices for $S_N$ and $E_N$, the algorithm runs in a BFS manner to preserve the topological characteristics of the graph and only chooses those candidates that are already connected to $T_D$. After successful vertices selection, the link is inserted between $S_N$ and $E_N$, this way single inserted link fulfills the requirement of two vertices. New link insertion changes the degrees of candidate vertices, thus *conditionset*(*degree*, *value*) and *DSeq* are updated accordingly. The algorithm runs until every element in *DSeq* appears at least *k* times. After the anonymization process, there might be a possibility for *DSeq* to contain an element $E(DoV)$, which remains un-anonymized even after the whole degree anonymization process. In such a scenario, *E* and its associated links will be removed from *DSeq*. Link deletion depends on *DSeq*, and might be performed only once.

---

**Algorithm 2:** Degree anonymization

---

1: DSeq implementation
2: **while** DSeq ! = empty **do**
3:     compute $C_{D(E)}$, $T_D - C_{D(E)}$ for each partition
4:     create condition_set(degree,value)
5:     key $\leftarrow C_{D(E)}$
6:     value $\leftarrow T_D - C_D$
7: **end while**
8: **while** condition_set ! = empty && ($S_N$ or $E_N$ = null) **do**
9:     **if** $S_N$ = null **then**
10:         choose first pair(degree,value)
11:         $S_N \leftarrow$ key
12:         $R_{L(S)} \leftarrow$ value
13:         **if** $S_N$ = only link **then**
14:             choose $S_N$
15:         **else**
16:             choose random $S_N$ connected to $T_D$
17:         **end if**
18:     **else**
19:         choose pair(degree,value) with same $R_{L(S)}$
20:         $E_N \leftarrow$ key
21:         $R_{L(E)} \leftarrow$ value
22:         **if** $E_N$ = only link **then**
23:             choose $E_N$
24:         **else**
25:             choose random $E_N$ connected to $T_D$
26:         **end if**
27:     **end if**
28: **end while**
29: insert link from $S_N$ to $E_N$
30: update pair_list(key,value)
31: update DSeq

---

This algorithm takes the degree sequence, constructs an anonymized degree sequence, and outputs the graph with exactly this sequence. It runs iteratively, on each iteration it chooses one candidate vertex for link insertion. It also maintains the condition set for candidate vertices and after every addition, this set is updated. For *L* links, the condition set has $O(L)$ entries. For choosing $S_N$ and $E_N$ from the condition set, it takes $O(L|V)$

time as it computes required links for each vertex. So, total running time of a proposed algorithm is $O(L^2|V|)$.

4.1.3. Privacy Analysis

In this section, the paper focuses on the analysis of the proposed *k*-NDDP approach in terms of privacy. The proposed approach modifies the original graph using the graph modification technique. Furthermore, for an attacker to determine whether the dummy links already exist or not in the original graph requires some confidence *C*.

A graph having $|V|$ as the total no of available vertices. It contains at most $|L_f|$ links. Average confidence of adversary for a link to existing in anonymized graph $G_a$ will be equal to the probability mentioned below, where $p_a$ and $p_d$ denotes added and deleted links, respectively.

$$C = \frac{|L_f| - |L|(1 - p_d + p_a)}{|L_f|}$$

DP is used to measure privacy by comparing the original graph and the published graph. Both neighboring graphs should have the following relationship for anonymization algorithm *Al* to achieve DP.

$$Pr[Al(G_1) = O] \leq e^\epsilon \times Pr[Al(G_2) = O]$$

This relationship suggests that the probability of a link to exist in $G_P$ is not greater than the probability $e^\epsilon$ does not exist.

**Theorem 1.** *For an attacker who holds the published graph, we assume that the probability of all its links existed (or did not exist) in the original graph is not higher than $e^\epsilon$ times the probability of all its links not existing (or existing) in the original graph. The value of $\epsilon$ is given by*

$$\epsilon = \ln\left(\frac{|L|.|L_f|(1 - p_d + p_a)}{|L_f|.|L| \cdot p_a} + \frac{|L_f| - |L|(1 - p_d + p_a)}{|L_f|} \times \frac{(|L_f| - |L|)(|L| \cdot p_d)}{(|L|)(|L_f| - |L| - |L| \cdot p_a)}\right)$$

**Proof.** Adversary chooses a pair of vertices $(a, b)$. This vertex pair has a link in one of the two neighboring graphs such that $G_1((a, b) \in L_1)$ contains the link and $G_2((a, b) \notin L_2)$ does not. After anonymization graphs are published, both publishing graphs $G_{p1}$ and $G_{p2}$ should have same output. There are two cases for this vertex pair $(a, b)$, which are $((a, b) \in L_p)$ and $((a, b) \notin L_p)$ shown in Table 2. First, consider the case when both published graphs contain a link between $(a, b)$ such that $O = (a, b) \in L_p$

$$Pr[Al(G_1) = O] = Pr[(a, b) \in L_{p1} | (a, b) \in L_1] \tag{1}$$

$$Pr[Al(G_2) = O] = Pr[(a, b) \in L_{p2} | (a, b) \notin L_2] \tag{2}$$

Simplifying Equations (1) and (2)

$$Pr[(a, b) \in L_{p1} | (a, b) \in L_1] = Pr\left(\frac{[(a, b) \in L_{p1} \cap (a, b) \in L_1]}{Pr[(a, b) \in L_1]}\right) = \frac{|L|(1 - p_d)}{|L|}$$

$$Pr[(a, b) \in L_{p2} | (a, b) \notin L_2] = Pr\left(\frac{[(a, b) \in L_{p2} \cap (a, b) \notin L_2]}{Pr[(a, b) \notin L_2]}\right) = \frac{|L| \cdot p_a}{|L_f| - |L|}$$

$$\epsilon_1 = \frac{Pr[Al(G_1) = O]}{Pr[Al(G_2) = O]} = \frac{(|L_f| - |L|)(1 - |L| \cdot p_a)}{|L| \cdot p_a} \tag{3}$$

Now, consider the second case when both published graphs do not contain a link between $O = (a, b) \notin L_p$. Similarly, by simplifying Equation (1) and (2)

$$Pr[Al(G_1) = O] = Pr[(a, b) \notin L_{p1}|(a, b) \in L_1] = \frac{|L| \cdot p_d}{|L|}$$

$$Pr[Al(G_2) = O] = Pr[(a, b) \notin L_{p2}|(a, b) \notin L_2] = \frac{|L_f| - |L| - |L| \cdot p_a}{|L_f| - |L|}$$

$$\epsilon_2 = \frac{(|L_f| - |L|)(|L| \cdot p_d}{(|L|)(|L_f| - |L| - |L| \cdot p_a)} \tag{4}$$

From the probabilities of first and second case shown in Equation (3) and (4), respectively, the average privacy of $\epsilon$ is achieved

$$\epsilon = \frac{(|L|(1 - p_d + p_a)}{|L_f|}\epsilon_1 + \frac{|L_f| - |L|(1 - p_d + p_a)}{|L_f|}\epsilon_2$$

Hence, it is proved.

$$\epsilon = \ln\left(\frac{|L| \cdot |L_f|(1 - p_d + p_a)}{|L_f|.|L| \cdot p_a} + \frac{|L_f| - |L|(1 - p_d + p_a)}{|L_f|} \times \frac{(|L_f| - |L|)(|L| \cdot p_d)}{(|L|)(|L_f| - |L| - |L| \cdot p_a)}\right)$$

□

**Table 2.** Links and non-links in $G$ and $G_p$.

|  | Links in $G$ | Non-Links in $G$ | Sum |
|---|---|---|---|
| Links in $G_p$ | $|L|(1 - p_d)$ | $|L| \cdot p_a$ | $|L|(1 - p_d + p_a)$ |
| Non-Links in $G_p$ | $|L| \cdot p_d$ | $|L_f| - |L| - |L| \cdot p_a$ | $|L_f| - |L|(1 - p_d + p_a)$ |
| Sum | $|L|$ | $|L_f| - |L|$ | $|L_f|$ |

## 5. Experiments

This section assess the working of *k*-NDDP algorithm on three real datasets to evaluate its privacy and utility. The utility is evaluated by the topological characteristics of the graph.

### 5.1. Datasets

To evaluate the utility of the proposed *k*-NDDP approach three real-world datasets are utilized. These datasets are undirected social graphs having one link between two vertices. Links on these social graphs represent friendships or neighbor connections of a vertex (user). Datasets are available on two different repositories [47,48]. SOCFB-USFCA72 is a social network dataset available at [49]. It is a Facebook extracted dataset containing 58,228 vertices and 214,078 connected links that represent the interaction between vertices. FEATHER-DEEZER-SOCIAL [50] represents vertices as users from European countries and links as mutual relationships between them. This dataset contains 28,281 vertices and 92,752 links. FEATHER-LASTFM-SOCIAL [50] is a social network dataset that represents vertices as users from Asian countries and links as relationships between them. It consists of 7624 vertices and 27,806 links.

### 5.2. Evaluation Metrics

The original social graph $G = (V, L)$ is converted into anonymized graph $G_a = (V_a, L_a)$ by applying the proposed degree anonymization scheme. To evaluate the effectiveness of the published graph, structural properties of the graph are examined. Structural properties of the graph include average shortest path length (APL) [51], clustering coefficient (CC) [52] and betweenness centrality (BW) [53].

- Average shortest path length: APL measures the efficiency of information that transport through the network. This concept calculates the mean path between two vertices, which is the average shortest path length of those vertices. APL calculates the mean path for all possible pairs of network vertices by using this formula.

$$APL = \frac{1}{p(p-1)} \sum_{i \neq j} d(n_1, n_2)$$

- Clustering coefficient: CC is a measure of the degree of vertices that make closer clusters with each other. This concept calculates the average clustering coefficient of all available vertices, which depends on locality

$$C_n = \frac{2L_n}{k_n(k_n - 1)}$$

The clustering coefficient for the whole graph is the average of the local values $C_n$.

$$C = \frac{1}{v} \sum_{n=1}^{m} C_n$$

- Betweenness centrality: BW is a measure of centrality of all vertices in the graph based on their shortest paths. It calculates the shortest path from all vertices that pass through one specific vertex. For vertex $V$, it is calculated using this formula.

$$G_n = \sum_{a \neq n \neq b} \frac{\sigma_{ab(n)}}{\sigma_{ab}}$$

Matrices listed above are widely used to evaluate the performance of graph publishing algorithms. The closer the value of the anonymized graph to the original graph, the more utility is maintained.

*5.3. Experimental Evaluation*

The utility of the proposed *k*-NDDP anonymization approach is inspected through the matrices listed above. The smaller difference between values of original data and achieved results, the more utility method preserves. Figure 7 represents the results of first experiment derived from different values of *k* for CC, APL and BW. For the first experiment, (FEATHER-DEEZER-SOCIAL) dataset is utilized, which is an undirected social graph. Figure 7a shows different values of CC for the original and anonymized graph. The constant line is the representation of the original value. After anonymization, the value of CC increases with increase of *k*, but the deviation between original and anonymized values is very little. The reason for this increase is the proposed methodology. The approach determines candidate vertices of link insertion, and a single inserted link fulfills the requirement of two vertices. This process makes density between the vertices and increases the value of CC. Figure 7b details the APL for the vertices of original and anonymized graphs. The horizontal constant line represents the APL of the original graph for different values of *k*. The value of APL, under the proposed *k*-NDDP approach, decreases with the increase of privacy parameter *k* as algorithm preserves shortest path information. Figure 7c shows the BW of the anonymized and original graph. The larger the value of BW, the more vertices dominate the graph. This indicates that the vertices with high BW have more control over the graph. After anonymization, the value of BW decreases, but the difference between original and anonymized results is very small. The results of second experiment for FEATHER-LASTFM-SOCIAL are shown on Table 3.

**Table 3.** Properties of *k*-NDDP anonymized social network dataset (FEATHER-LASTFM-SOCIAL).

| *k* | APL | CC | BW |
|---|---|---|---|
| 0 | 2.65 | 0.0040 | 0.015 |
| 5 | 2.64 | 0.0041 | 0.014 |
| 10 | 2.63 | 0.0043 | 0.013 |
| 15 | 2.62 | 0.0045 | 0.012 |
| 20 | 2.60 | 0.0048 | 0.011 |
| 25 | 2.60 | 0.0049 | 0.010 |

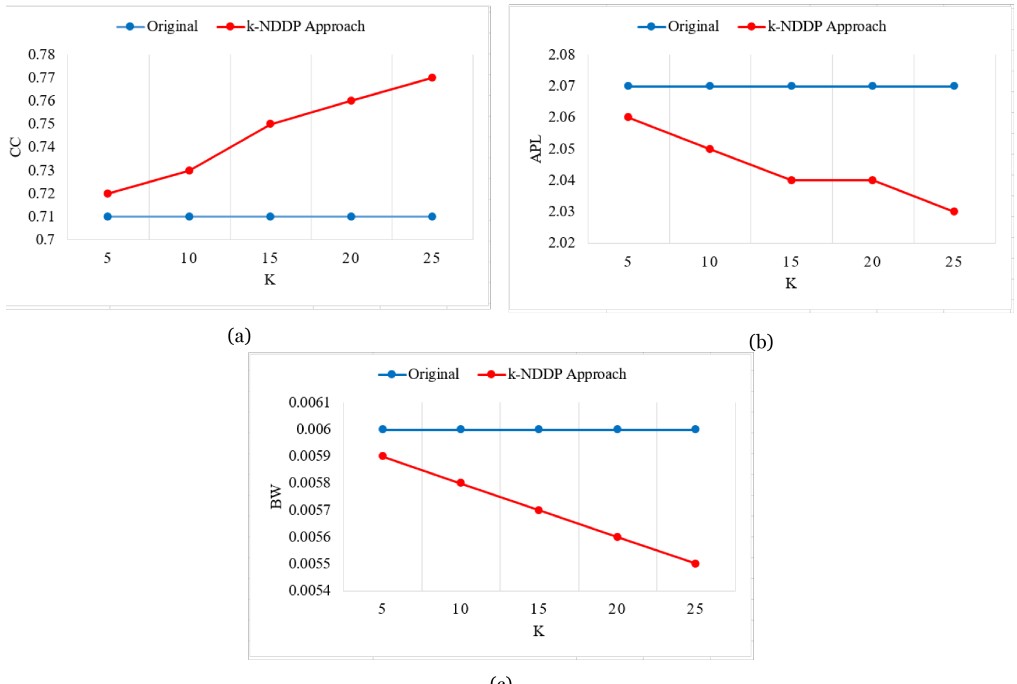

**Figure 7.** Comparison of *k*-NDDP with *k*-NMF(FEATHER-DEEZER-SOCIAL). (**a**) CC (**b**) APL (**c**) BW.

Now, for last experiment, SOCFB-USFCA72 dataset is utilized and its utility preservation with *k*-NMF is evaluated. The difference in CC for both approaches is shown in Figure 8a. After anonymization, the value of CC increases for both approaches. The original value of CC is 0.0267, and *k*-NDDP obtains 0.0272, while *k*-NMF obtains 0.0275 average value. Figure 8b represents the result of APL. For every value of *k*, *k*-NDDP preserves shortest path information, thus the value of APL comparatively decreases more in this approach. The actual value of APL for original dataset is 1.3715, after anonymization *k*-NDDP and *k*-NMF achieves 1.3711 and 1.3709 average value, respectively. BFS traversal for candidate vertex selection preserves the shortest path between two vertices and shows similar performance as *k*-NMF in terms of betweenness centrality shown in Figure 8c, as there is a negligible change between different values for *k* for both approaches (0.00056 and 0.00057). These results show that for different values of *k*, the divergence in CC, BW, and APL is not more than 0.8%.

Figure 9 represents the running time comparison and change in edges for both approaches. Anonymization time of *k*-NDDP is less than *k*-NMF and inserted links are 60% less, as compared to *k*-NMF approach. This experiment concludes that proposed approach is better in terms of noise addition as it adds fewer dummy links to the graph and performs well as compared to *k*-NMF. *k*-NMF takes $O(L^2, V^2)$ time to execute the anonymization algorithm, and the proposed approach takes a total $O(L^2, V)$ time to complete. Hence, it concludes that *k*-NDDP is more suitable for adding a minimum number of links using minimum steps with less time complexity. In this way, *k*-NDDP provides better privacy protection from identity disclosures while maintaining utility.

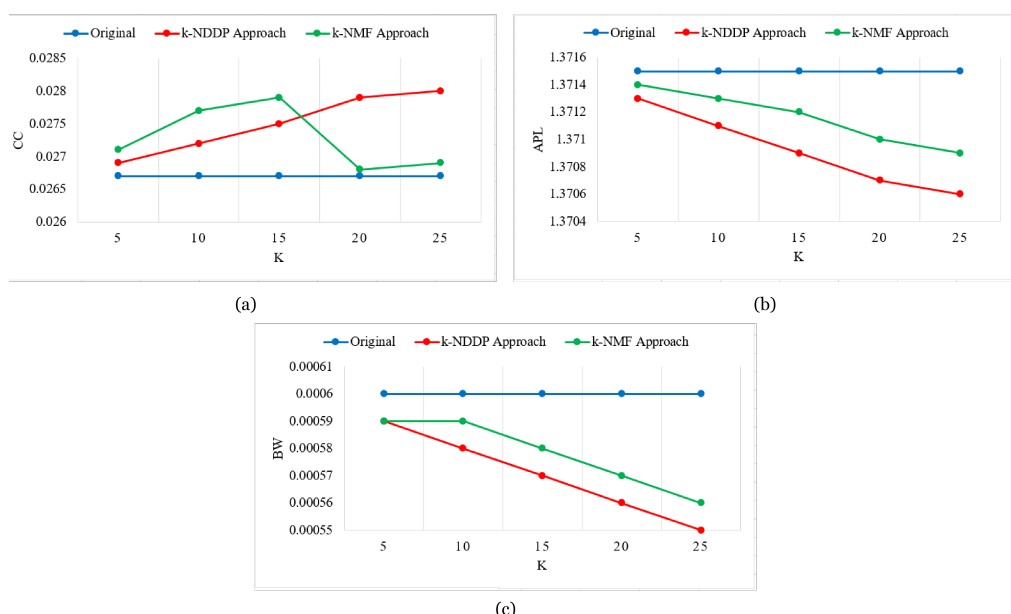

**Figure 8.** Comparison of *k*-NDDP with *k*-NMF(SOCFB-USFCA72). (**a**) CC; (**b**) APL; (**c**) BW.

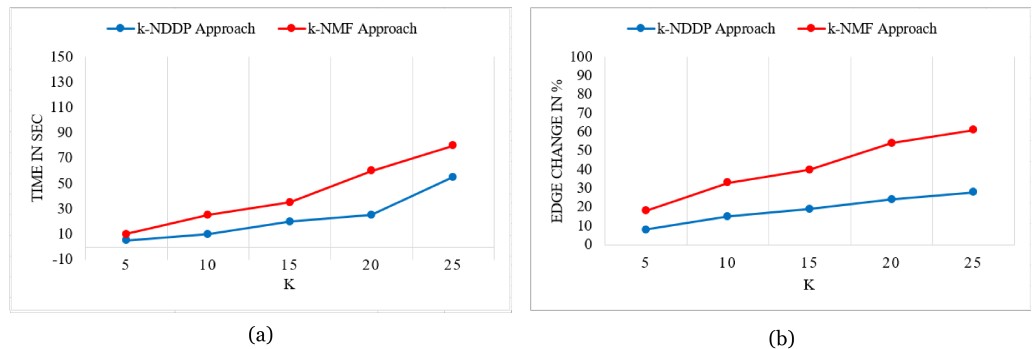

**Figure 9.** Comparison of *k*-NDDP with *k*-NMF(SOCFB-USFCA72). (**a**) Time; (**b**) edge change.

## 6. Conclusions

This paper proposed a novel approach *k*-NDDP, which is the extension of *k*-NMF. *k*-NDDP is a degree anonymization method that extends the concept of *k*-anonymity and differential privacy based on Node DP for vertex degrees. The proposed approach provides a solution to the problem that reveals the individual behind any vertex of the social graph and causes identity disclosure. To defend against identity disclosures, the suggested method inserts the least number of dummy connections into the original graph while preventing the adversary from identifying the vertices and preserving as much graph information as possible. This scheme uses BFS traversal for candidate vertex selection during the link addition process that helps to preserve the maximum structural properties of the graph. The proposed approach implements a condition set with the *key* and *value* arguments that accumulate the degree and required links for the vertices. A single inserted link meets the requirements of other vertices during the anonymization process. The experimental evaluations showed the effectiveness of the proposed model in terms of utility, and privacy analysis proves that the suggested model is secure against identity disclosures.

**Author Contributions:** Supervision: A.A. (Adeel Anjum) and A.A. (Alia Asheralieva); writing—original draft: S.S.; writing—review: M.A. and and A.A. (Alia Asheralieva); writing—review and editing: S.S. All authors have read and agreed to the published version of the manuscript.

**Funding:** This work was supported in part by the National Natural Science Foundation of China (NSFC) Project No. 61950410603.

**Data Availability Statement:** We have used three datasets. SOCFB-USFCA72 is a social network dataset available at http://snap.stanford.edu/data (accessed on 19 May 2021). FEATHER-DEEZER-SOCIAL and FEATHER-LASTFM-SOCIAL datasets are available on https://paperswithcode.com/dataset/reddit (accessed on 19 May 2021).

**Conflicts of Interest:** The authors declare no conflict of interest.

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
