# Peer review of "k-NDDP: An Efficient Anonymization Model for Social Network Data Release"

_electronics, doi:10.3390/electronics10192440_

Round 1
Reviewer 1 Report
This is a good article in general. Well written. Well presented. The state of the art has been covered in a satisfactory manner. The methodology is clear. The results are available.
My only remark concerns the references which are a bit old and need to be updated with new and good quality references. This has an impact on the discussion as well, which should take this into account.
Author Response
RC: This is a good article in general. Well written. Well presented. The state of the art has been covered in a
satisfactory manner. The methodology is clear. The results are available. My only remark concerns the
references which are a bit old and need to be updated with new and good quality references. This has an
impact on the discussion as well, which should take this into account.
AR: Thank you for noting this. As you suggested, we updated the reference section and included some new
references in our manuscript. New added references are [6], [12], [7], [23], [1], [5], [13], [18], [14], [19], [15],
[20], [22], [8], [25], [11], [10], [21], [24], [16], [6], [2], [17], [9], [3] and [4]. (Page 17 & 18 of manuscript).
References
[1] Safia Bourahla, Maryline Laurent, and Yacine Challal. “Privacy preservation for social networks
sequential publishing”. In: Computer Networks 170 (2020), p. 107106.
[2] Yumeng Fu et al. “Privacy Preserving Social Network Against Dopv Attacks”. In: International
Conference on Web Information Systems Engineering. Springer. 2018, pp. 178–188.
[3] Tianchong Gao and Feng Li. “Protecting Social Network With Differential Privacy Under Novel Graph
Model”. In: IEEE Access 8 (2020), pp. 185276–185289.
[4] Tianchong Gao and Feng Li. “Sharing social networks using a novel differentially private graph model”.
In: 2019 16th IEEE Annual Consumer Communications & Networking Conference (CCNC). IEEE.
2019, pp. 1–4.
[5] Jiafeng Hu. “Effective and efficient algorithms for large graph analysis”. In: HKU Theses Online
(HKUTO) (2018).
[6] Haiping Huang et al. “Privacy-preserving approach PBCN in social network with differential privacy”.
In: IEEE Transactions on Network and Service Management 17.2 (2020), pp. 931–945.
[7] M Kiranmayi and N Maheswari. “A Review on Privacy Preservation of Social Networks Using Graphs”.
In: Journal of Applied Security Research (2020), pp. 1–34.
[8] Yang Li and Xuhua Hu. “Social network analysis of law information privacy protection of cybersecurity
based on rough set theory”. In: Library Hi Tech (2019).
[9] Zehua Lin. “Privacy Preserving Social Network Data Publishing”. PhD thesis. Miami University, 2021.
[10] Alex X Liu and Rui Li. “Publishing Social Network Data with Privacy Guarantees”. In: Algorithms for
Data and Computation Privacy. Springer, 2021, pp. 279–311.
[11] Baida Ouafae et al. “Data Anonymization in Social Networks”. In: (2020).
[12] Manos Papoutsakis et al. “Towards a Collection of Security and Privacy Patterns”. In: Applied Sciences
11.4 (2021), p. 1396.
[13] Ruggero G Pensa, Gianpiero Di Blasi, and Livio Bioglio. “Network-aware privacy risk estimation in
online social networks”. In: Social Network Analysis and Mining 9.1 (2019), pp. 1–15.
[14] Vu Viet Hoang Pham et al. “Privacy issues in social networks and analysis: a comprehensive survey”.
In: IET networks 7.2 (2018), pp. 74–84.
[15] Gu Qiuyang et al. “Dynamic social privacy protection based on graph mode partition in complex social
network”. In: Personal and Ubiquitous Computing 23.3 (2019), pp. 511–519.
1
[16] Youyang Qu et al. “Customizable Reliable Privacy-Preserving Data Sharing in Cyber-Physical Social
Network”. In: IEEE Transactions on Network Science and Engineering (2020).
[17] Abdullah Abdulabbas Nahi Al-Rabeeah and Mohammed Mahdi Hashim. “Social Network Privacy
Models”. In: Cihan University-Erbil Scientific Journal 3.2 (2019), pp. 92–101.
[18] Nemi Chandra Rathore and Somanath Tripathy. “InfoRest: Restricting Privacy Leakage to Online
Social Network App”. In: arXiv preprint arXiv:1905.06403 (2019).
[19] Khondker Jahid Reza, Md Zahidul Islam, and Vladimir Estivill-Castro. “Privacy Preservation of Social
Network Users Against Attribute Inference Attacks via Malicious Data Mining.” In: ICISSP. 2019,
pp. 412–420.
[20] AL-Kharji Sarah, Yuan Tian, and Mznah Al-Rodhaan. “A Novel (K, X)-isomorphism Method for
Protecting Privacy in Weighted social Network”. In: 2018 21st Saudi Computer Society National
Computer Conference (NCC). IEEE. 2018, pp. 1–6.
[21] Julian Steil et al. “Privacy-aware eye tracking using differential privacy”. In: Proceedings of the 11th
ACM Symposium on Eye Tracking Research & Applications. 2019, pp. 1–9.
[22] Marcin Waniek et al. “Hiding individuals and communities in a social network”. In: Nature Human
Behaviour 2.2 (2018), pp. 139–147.
[23] Xingping Xian et al. “Towards link inference attack against network structure perturbation”. In:
Knowledge-Based Systems (2021), p. 106674.
[24] Min Ye and Alexander Barg. “Optimal schemes for discrete distribution estimation under locally
differential privacy”. In: IEEE Transactions on Information Theory 64.8 (2018), pp. 5662–5676.
[25] Jinquan Zhang et al. “RcDT: Privacy preservation based on R-constrained dummy trajectory in mobile
social networks”. In: IEEE Access 7 (2019), pp. 90476–90486.

Reviewer 2 Report
My only recommendation is to add more social networks names beside Facebook (or is the worst?) or not to mention Facebook, but keep general to social networks.
Also, the experiments were done only with Facebook datasets? why?
Author Response
RC: My only recommendation is to add more social networks names beside Facebook (or is the worst?) or
not to mention Facebook, but keep general to social networks. Also, the experiments were done only with
Facebook datasets? Why?
AR: We appreciate your insightful suggestions. There was not any specific reason behind mentioning Facebook. It
was just a general sentence describing popularity reasons of social sites. So, as you suggested we made it
general for social networks shown as "Because of this popularity, people use social sites to connect with their
friends and family, share their interests, and establish connections". (Page 1 of manuscript).
1. Thank you for your question. But we want to clear this misconception. We used three different datasets
for our experiments. (Page 13 of manuscript).
• SOCFB-USFCA72 (Facebook extracted dataset).
• FEATHER-DEEZER-SOCIAL (Social network of Deezer users from Europe).
• FEATHER-LASTFM-SOCIAL (Social network of LastFM users from Asia).

Reviewer 3 Report
Review report on manuscript number ID electronics-1246682 submitted to "electronics".
Title: k-NDDP: An efficient anonymization model for social network data release
Authors: Shafaq Shakeel, Adeel Anjum, Alia Asheralieva and Masoom Alam
The subject is interesting and relevant to the field of this journal. On the other hand, the paper has some key omissions that have to be corrected before it is suitable for publication.
Moderate English changes required.
Τhe abstract does not provide the reader with information about the results. The authors mention only “The experiment shows that our approach protects the privacy of individuals on published graphs and maintains data utility and information loss effectively”. This sentence is too general. The abstract needs to be improved at this point.
The author should change all the sentences in the text which are written in the first plural.
In Section 5 the authors present the results of the proposed method. The results are not presented in an understandable way. They are not explained enough. This section should be rewritten more analytically.
There is no mention of Table 3 in the text. Please comment on the results of the table in the text.
The conclusions are too general. The authors should rewrite more analytically this section giving more information about the innovation of this work.
Author Response
RC: The subject is interesting and relevant to the field of this journal. On the other hand, the paper has some
key omissions that have to be corrected before it is suitable for publication.
1. Moderate English changes required.
2. The abstract does not provide the reader with information about the results. The authors mention
only “The experiment shows that our approach protects the privacy of individuals on published
graphs and maintains data utility and information loss effectively”. This sentence is too general.
The abstract needs to be improved at this point.
3. The author should change all the sentences in the text which are written in the first plural.
4. In Section 5 the authors present the results of the proposed method. The results are not presented
in an understandable way. They are not explained enough. This section should be rewritten more
analytically.
5. There is no mention of Table 3 in the text. Please comment on the results of the table in the text.
6. The conclusions are too general. The authors should rewrite more analytically this section giving
more information about the innovation of this work.
AR: We sincerely appreciate your insightful comments and suggestions for revising the paper.
1. As you suggested, we did amendments in the revised manuscript.
2. We have corrected the mentioned sentence of abstract section in the revision. "An extensive empirical
study shows that for different values of k the divergence produced by k-NDDP for CC, BW and APL is
not more than 0.8% also added dummy links are 60% less, as compared to k-NMF approach, thereby it
validates that proposed k-NDDP approach provides strong privacy while maintaining the usefulness of
data". (Page 1 of manuscript).
3. All sentences of first plural are carefully changed such as "In this paper, the concept of k-NDDP
is presented which provides defense from identity disclosure that intrudes privacy of individual by
discerning its position on social graph. Our key contributions of this paper are following:". (Page 4 of
manuscript).
4. Results section of manuscript are rewritten in understandable way. "Utility of proposed k-NDDP
anonymization approach is inspected through the matrices listed above. The smaller difference between
values of original data and achieved results, the more utility method preserves. Figure 7 represents
the results of first experiment derived from different values of k for CC, APL and BW. For first
experiment, (FEATHER-DEEZER-SOCIAL) dataset is utilized which is an undirected social graph.
Figure 7a shows different values of CC for the original and anonymized graph. The constant line is the
representation of the original value. After anonymization,the value of CC increases with increase of
k but the deviation between original and anonymized values is very little....". (Page 14, 15 and 16 of
manuscript).
5. Thank you for noting this. We mentioned the Table 3 in text too. "The results of second experiment for
FEATHER-LASTFM-SOCIAL are shown on Table 3.". (Page 14 of manuscript).
3
6. The conclusion section is rewritten analytically in the revised manuscript. "This paper proposed a novel
approach k-NDDP which is the extension of k-NMF. k-NDDP is a degree anonymization method, that
extends the concept of k-anonymity and differential privacy based on Node DP for vertex degrees. The
proposed approach provides a solution to the problem that reveals the individual behind any vertex of
the social graph and causes identity disclosure. To defense against identity disclosures the suggested
method inserts the least number of dummy connections into the original graph while preventing the
adversary from identifying the vertices and preserving as much graph information as possible. This
scheme uses BFS traversal for candidate vertex selection during the link addition process that helps
to preserve the maximum structural properties of the graph. The proposed approach implements a
condition set with the key and value arguments that accumulate the degree and required links for the
vertices. A single inserted link meets the requirements of other vertices during the anonymization
process. The experimental evaluations showed the effectiveness of the proposed model in terms of
utility and privacy analysis proves that the suggested model is secure against identity disclosures".
(Page 16 & 17 of manuscript).
